# The Reissner fiber under tension in vivo shows dynamic interaction with ciliated cells contacting the cerebrospinal fluid

Celine Bellegarda[1], Guillaume Zavard[1], Lionel Moisan[2], Françoise Brochard-Wyart[3], Jean-François Joanny[3,4], Ryan S Gray[5], Yasmine Cantaut-Belarif[1], Claire Wyart[1]*†

[1]Sorbonne Université, Paris Brain Institute (Institut du Cerveau, ICM), Institut National de la Santé et de la Recherche Médicale U1127, Centre National de la Recherche Scientifique Unité Mixte de Recherche 7225, Assistance Publique–Hôpitaux de Paris, Campus Hospitalier Pitié-Salpêtrière, Paris, France; [2]Université Paris Cité, CNRS, MAP5, Paris, France; [3]Paris Sciences et Lettres (PSL) University, Institut Curie, Sorbonne Université, Paris, France; [4]Paris Sciences et Lettres (PSL) University, Collège de France, Paris, France; [5]Dell Pediatrics Research Institute, The University of Texas at Austin, Austin, United States

*For correspondence:
claire.wyart@icm-institute.org

Present address: †Institut du Cerveau, Campus Hospitalier Pitié-Salpêtrière, Paris, France

## Abstract

The Reissner fiber (RF) is an acellular thread positioned in the midline of the central canal that aggregates thanks to the beating of numerous cilia from ependymal radial glial cells (ERGs) generating flow in the central canal of the spinal cord. RF together with cerebrospinal fluid (CSF)-contacting neurons (CSF-cNs) form an axial sensory system detecting curvature. How RF, CSF-cNs and the multitude of motile cilia from ERGs interact in vivo appears critical for maintenance of RF and sensory functions of CSF-cNs to keep a straight body axis, but is not well-understood. Using in vivo imaging in larval zebrafish, we show that RF is under tension and resonates dorsoventrally. Focal RF ablations trigger retraction and relaxation of the fiber's cut ends, with larger retraction speeds for rostral ablations. We built a mechanical model that estimates RF stress diffusion coefficient D at 5 mm²/s and reveals that tension builds up rostrally along the fiber. After RF ablation, spontaneous CSF-cN activity decreased and ciliary motility changed, suggesting physical interactions between RF and cilia projecting into the central canal. We observed that motile cilia were caudally-tilted and frequently interacted with RF. We propose that the numerous ependymal motile monocilia contribute to RF's heterogenous tension via weak interactions. Our work demonstrates that under tension, the Reissner fiber dynamically interacts with motile cilia generating CSF flow and spinal sensory neurons.

## Editor's evaluation

This exceptional work substantially advances our understanding of the mechanics of the Reissner's fibre (RF) by performing in-vivo experiments that track and analyze the behavior of the RF when it is cut and the behavior of ciliated cells touching the RF when contact is interrupted. The data is valuable and the conclusions are compelling. The work will be of broad interest to many research communities including developmental neuroscience and cilia biology.

## Introduction

The cerebrospinal fluid (CSF) is secreted by the choroid plexuses and fills the intracerebral ventricles, spinal and brain subarachnoid spaces, and the central canal of the spinal cord (*Lun et al., 2015*;

*Battal et al., 2011*). In the last decade, the physico-chemical properties of the CSF have been shown to impact the development of the nervous system by influencing neurogenesis (*Lehtinen et al., 2011*), neuronal migration (*Paul et al., 2017*) as well as morphogenesis (*Bearce and Grimes, 2021*). In zebrafish, the circulation of CSF modulates the geometry of the body axis during embryogenesis (*Zhang et al., 2018*) and spine curvature in juvenile/adult zebrafish (*Grimes et al., 2016*). Genetic investigations in zebrafish revealed that straight body axis formation and spine organogenesis rely on the expression of urotensin neuropeptides in the spinal cord by ciliated sensory neurons surrounding the central canal, referred to as CSF-contacting neurons (CSF-cNs) (*Quan et al., 2015*; *Zhang et al., 2018*; *Bearce et al., 2022*; *Gaillard et al., 2023*).

Spinal CSF-cNs were identified by Kolmer and Agduhr in over a hundred vertebrate species (*Kolmer, 1921*; *Agduhr, 1922*; *Vigh and Vigh-Teichmann, 1998*) as GABAergic sensory neurons that project an apical extension into the lumen of the central canal and extend an ascending axon into the spinal cord. The apical extension comprises one motile cilium and a brush of microvilli that bathe in the CSF (*Vigh and Vigh-Teichmann, 1998*). During development, CSF-cNs differentiate into dorso-lateral and ventral populations that are distinguished by distinct developmental origins (*Park et al., 2004*; *Huang et al., 2012*; *Petracca et al., 2016*). These two populations differ by the morphology of their ciliated apical extension and axonal projections, their expression of peptides (*Quan et al., 2015*; *Djenoune et al., 2017*; *Desban et al., 2019*; *Prendergast et al., 2023*) as well as the neuronal targets contacted by the CSF-cN axons (*Djenoune et al., 2017*; *Desban et al., 2019*; *Orts-Del'Immagine et al., 2014*).

CSF-cNs express urotensin-related peptides under the control of the Reissner Fiber (RF) (*Quan et al., 2015*; *Zhang et al., 2018*; *Cantaut-Belarif et al., 2020*; *Lu et al., 2020*), an acellular thread bathing in CSF within the lumen of the central canal and in close vicinity to the apical extension of CSF-cNs. Thanks to the beating of the numerous motile monocilia from ependymal radial glia, RF forms during development from the aggregation of the monomer SCO-spondin, a very large glyco-protein (*Sterba et al., 1982*; *Cantaut-Belarif et al., 2018*; *Troutwine et al., 2020*). SCO-spondin is initially secreted into the CSF by cells in the floor plate, the flexural organ and the subcommissural organ (SCO), and is later solely produced by the SCO (*Meiniel and Meiniel, 2007*; *Gobron et al., 1999*). In larval zebrafish, RF diameter is approximately 200 nm (*Orts-Del'Immagine et al., 2020*) and RF displays slow, continual rostrocaudal movement (*Troutwine et al., 2020*). In mutant embryos for scospondin in which RF does not form, the body axis is curled down (*Cantaut-Belarif et al., 2018*). Furthermore, in scospondin mutants in which RF forms in the embryo but is not maintained in juvenile stages (*Troutwine et al., 2020*; *Rose et al., 2020*), the body axis becomes straight during embryogenesis, but later on at the juvenile stage the spine undergoes 3D torsion, a hallmark of idiopathic scoliosis. Peptides from the family of urotensin (Urp1, Urp2) and their receptor Uts2r3 play a major role in straightening the body axis throughout life (*Bearce et al., 2022*; *Gaillard et al., 2023*). Remarkably, urotensin signaling is relevant in human patients with adolescent idiopathic scoliosis (*Dai et al., 2021*; *Xie et al., 2023*).

We previously showed that CSF-cNs sense zebrafish body curvature in vivo (*Böhm et al., 2016*). Accordingly, CSF-cNs can respond in vitro to pressure-application in an open book preparation of the lamprey spinal cord (*Jalalvand et al., 2016*) as well as to mechanical stimulation applied to CSF-cN cell membranes in primary cultures (*Sternberg et al., 2018*). In zebrafish, CSF-cNs sense spinal curvature in vivo only on the concave side (*Böhm et al., 2016*), a process that requires intact Pkd2l1 channels. This process relies on a modulation of the opening probability of the mechanosensory channel Pkd2l1 (*Sternberg et al., 2018*) for CSF-cNs on the concave side, ipsilateral to where skeletal muscles contract (*Böhm et al., 2016*). In zebrafish lacking RF, both the CSF flow (*Cantaut-Belarif et al., 2018*) and Pkd2l1 channel activity (*Sternberg et al., 2018*) are unaffected, yet CSF-cN response to spinal curvature was reduced (*Orts-Del'Immagine et al., 2020*). These observations suggest that an intact RF amplifies the mechanosensitivity of CSF-cNs, either by RF contacting the ciliated CSF-cN apical extension or by enhancing the gradient of CSF flow that CSF-cNs are subject to during spinal bending. Previous investigations on the structure of RF and its interaction with CSF-cNs were performed in fixed tissues (using 4% paraformaldehyde; see *Cantaut-Belarif et al., 2018*; *Orts-Del'Immagine et al., 2020*), which leads to deformation of the central canal that can alter the relative distance of the RF to CSF-cNs (*Orts-Del'Immagine et al., 2020*; *Thouvenin et al., 2020*). It is therefore not yet clear how CSF-cN activity is affected by interacting with RF in vivo.

In this study, we took advantage of the transparency of transgenic zebrafish larvae to investigate in vivo the dynamic properties of RF and its interaction with ciliated cells contacting the CSF in the central canal. By developing a tracking method to identify the dorsoventral position of RF along the rostrocaudal axis, we discovered that, at the larval stage, RF is under tension and undergoes dynamic spontaneous oscillatory activity in vivo over 100 – 200 nm in the dorsoventral axis, with largest amplitudes of oscillations in the middle portion of the fish. Interestingly, the focal ablation of RF using a pulsed laser led to an initial fast retraction of RF with maximal retraction speeds reaching up to 700 µm/s when the ablation was performed rostrally. After an initial retraction, relaxation of RF was sometimes accompanied by a retention of the cut end in the central canal with an observable deformation of the fiber when it seemed to interact with other cellular components in the CSF. We built a mechanical model of RF to estimate from the relaxation kinetics its elastic properties, including its mechanical diffusion coefficient, characteristic time, and retraction ratio. Our model revealed a heterogeneous tension along RF that was larger on the rostral portion, which is not likely explained by CSF flow. In order to investigate whether motile cilia, including the most numerous ones from ependymal radial glial cells (ERGs) and the sparse ones from CSF-cNs, were interacting with RF, we performed acute focal ablations of RF using pulsed lasers. We found that RF ablation decreased spontaneous calcium activity in CSF-cNs. We observed as well frequent interactions between RF and the tip of beating monocilia. Accordingly, we found that the frequency of ciliary beating was often affected by RF ablation. Together with our observations that the polarity of cilia beating in the sagittal plane was biased caudally, our findings suggest that caudally-oriented beating of monocilia may contribute to the generation of an heterogeneous tension via weak interactions between motile cilia and the fiber. Altogether, our observations indicate that RF is a dynamic structure under tension in the CSF that interacts with CSF-cNs as well as with the motile monocilia in the central canal. The dynamic RF subsequently promotes activity in a subset of sensory neurons at rest, while the implications of the interactions between RF and long motile cilia from ependymal radial glia remain to be investigated.

## Results

### The Reissner fiber under tension spontaneously oscillates in vivo

To investigate the dynamical properties of the Reissner fiber, we performed high-speed imaging of the central canal in the transgenic reporter knock-in zebrafish line *Tg(sspo:sspo-GFP)* (*Troutwine et al., 2020*), in which the fiber is GFP-tagged (*Figure 1*). In 3 days post fertilization (dpf) *Tg(sspo:sspo-GFP)* larvae after paralysis (see Materials and methods), we observed that RF was straight and taut in the sagittal plane and rapidly changed position along the dorsoventral axis over time in the central canal (*Figure 1A and B*). These observations indicate that the fiber is under tension in vivo. In contrast, RF of larvae after fixation was slack and stationary (*Figure 1C*; see also *Video 1* in paralyzed live larvae, *Video 2* after fixation, *Video 3* in unparalyzed live larvae). We developed a script to track the position of the fiber in the dorsoventral axis (*Figure 1D*; see also Materials and methods) as a function of its position in the rostrocaudal axis, discretized in 2 µm bins (*Figure 1E1 and E2*). We estimated that the dorsoventral displacement of the fiber from its mean position in the central canal (on average median ± standard deviation for all values provided hereafter: 74 nm ± 68 nm) was significantly larger in paralyzed living larvae than in euthanized larvae after fixation (on average: 32 nm ± 39 nm; unpaired two-tailed t-test: $p < 10^{-4}$; *Figure 1F1*), whose perceived displacement may be due to noise in our imaging setup and artifacts of detection of the center position of the fiber. The amplitude of RF dorsoventral displacement was largest in the middle portion (125 nm ± 108 nm), followed by the fiber displacement on the rostral side (100 nm ± 103 nm; Tukey's HSD Test for multiple comparisons: $p < 10^{-4}$; *Figure 1F1*). The median of the amplitudes of RF displacement in the caudal end (70 nm ± 68 nm; Tukey's HSD Test for multiple comparisons: $p < 10^{-4}$; *Figure 1F2*) was the closest to the fiber displacement of fixed larvae, possibly partly reflecting that the fiber is anchored on both the rostral and caudal ends of the fish. Recordings from rostral, middle and caudal portions of different fish showed a greater variability in dorsoventral displacement from the mean position in the middle portion of the fish than on the rostral and caudal portions (*Figure 1F3*). Performing a spatial principal component analysis over all pixels in the video revealed that the first two components after dimensionality reduction represented dorsoventral translation and local rotation of the fiber, respectively (*Figure 1G*; see

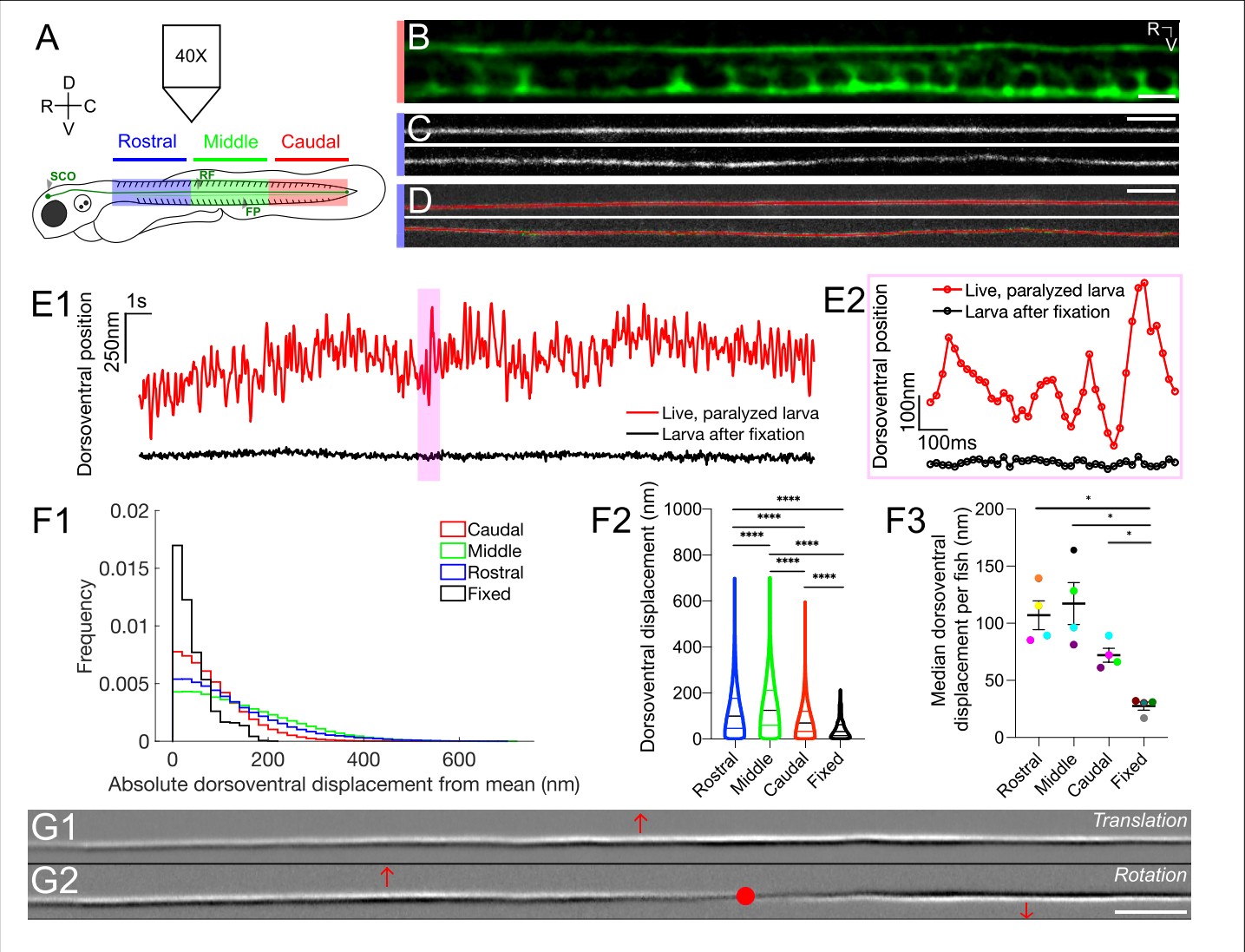

**Figure 1.** The Reissner fiber under tension exhibits spontaneously dynamic behavior over the dorsoventral axis in the central canal. (**A**) Spinning disk confocal microscopy setup using a 40X objective with 3 dpf *Tg(sspo:sspo-GFP)* zebrafish larvae for live imaging. Schema of zebrafish larva designates rostral (blue), middle (green) and caudal (red) sections, corresponding to somites 1–10, 11–20, 21–30, respectively. (**B**) Immunohistochemistry with anti-GFP antibodies in 3 dpf *Tg(sspo:sspo-GFP)* larva after fixation shows RF with the floor plate (FP) visible within the caudal somites of the spinal cord. (**C**) Live imaging snapshot of RF in the rostral somites of a 3 dpf *Tg(sspo:sspo-GFP)* paralyzed, living larva (top) and larva after fixation (bottom). (**D**) Example tracking of continuous motion of RF through the development of a script to model its movements in the dorsoventral axis. (**E1**) Example trace of the change in dorsoventral position of RF in one paralyzed, living larva (red) and another euthanized larva after fixation (black) over a 25 s timelapse acquired at 40 Hz. Data was discretized in 2 µm bins along the rostrocaudal axis before plotting. (**E2**) Zoomed-in display of the highlighted area marked on E1, showing a trace of the dorsoventral position of RF over 1 s for both the paralyzed, living larva and the euthanized larva after fixation, respectively, with circles indicating the sampling points. (**F1**) Displacement in the dorsoventral axis for paralyzed larvae (N=4 rostral, 4 middle, 4 caudal recordings) is significantly larger than that of fixed (N=4 recordings) larvae on average median ± standard deviation provided hereafter: in paralyzed living larvae = 74 nm ± 68 nm versus in fixed larvae = 32 nm ± 39 nm; unpaired two-tailed t-test: $p < 10^{-4}$. The displacement was calculated from data that was discretized in 2 µm bins along the rostrocaudal axis. (**F2**) Dorsoventral displacement of RF is significantly different among rostral, middle and caudal segments of paralyzed larvae (on average median ± standard deviation in rostral somites = 100 nm ± 103 nm versus in middle somites = 125 nm ± 108 nm versus in caudal somites = 70 nm ± 68 nm versus in fixed larvae = 32 nm ± 39 nm; Tukey's HSD Test for multiple comparisons: $p < 10^{-4}$). (**F3**) Median dorsoventral displacement of RF from the mean position per fish, with each color representing a different fish (on average mean of median dorsoventral displacement ± standard deviation in rostral somites = 107 nm ± 25 nm versus in middle somites = 117 nm ± 37 nm versus in caudal somites = 72 nm ± 12 nm versus in fixed larvae = 28 nm ± 7 nm; Tukey's HSD Test for multiple comparisons: $p < 0.05$). (**G1–G2**) A principal component analysis was computed on one image sequence of one fish (with each image corresponding to one observation) to understand the most significant movements of the fiber. The G1 component corresponds to a dorsoventral translation (the fiber borders are black on the ventral side, white on the dorsal side) and G2 to a small local rotation around a point (in red). These two components account respectively for 21.9% (**G1**) and 3.4% (**G2**) of the total temporal variability in the video. * $p < 0.05$, **** $p < 10^{-4}$. Scale bar is 10 µm (**B, C, D**) and 20 µm (**G1, G2**).

**Video 1.** Time series showing the movement of the Reissner fiber in the sagittal plane of live *Tg(sspo:sspo-GFP)* transgenic larvae. Data was acquired at 40 Hz for 25 s. Rostral, left and dorsal, top. Video is replayed in real time (40 Hz). Scale bar represents 15 µm.
https://elifesciences.org/articles/86175/figures#video1

**Video 2.** Time series showing the lack of movement of the Reissner fiber in the sagittal plane in 3 dpf *Tg(sspo:sspo-GFP)* transgenic larvae after fixation. Data was acquired at 40 Hz for 25 s. Rostral, left and dorsal, top. Video is replayed in real time (40 Hz). Scale bar represents 15 µm.
https://elifesciences.org/articles/86175/figures#video2

also *Video 4*). These observations uncover that RF under tension in vivo demonstrates dynamic dorsoventral oscillations with graded amplitudes of oscillations along the rostrocaudal axis.

## The Reissner fiber enhances spontaneous calcium activity in cerebrospinal fluid-contacting neurons

Given that the fiber in paralyzed, living larvae exhibits spontaneous dorsoventral movements, we investigated whether these oscillations contribute to the spontaneous calcium activity of the CSF-cNs (*Figure 2*). We performed acute 2-photon ablations of RF and tested whether it impacted the spontaneous calcium activity of CSF-cNs in triple transgenic *Tg(sspo:sspo-GFP;pkd2l1:tagRFP;pkd2l1:GCaMP5G)* paralyzed larvae (*Figure 2A1 and A2*; see also *Video 5* before RF ablation, *Video 6* after RF ablation and Materials and methods). To record the spontaneous activity of CSF-cNs before and after RF photoablation, we monitored calcium transients of ventral CSF-cNs located in the sagittal plane of RF (*Figure 2B*). Overall, fewer cells were active after RF photoablation (on average 11% compared to 28%; paired two-tailed t-test: $p < 0.05$; *Figure 2C*), and calcium activity decreased by 45% on average across larvae (*Figure 2D*), with a subset of ventral CSF-cNs showing decreased activity after RF photoablation. The number of calcium events occurring within those cells decreased after RF ablation (*Figure 2E*, on average 0.94 events/min before and 0.87 events/min after in active cells; paired two-tailed t-test: $p < 0.005$). Altogether, our results indicate that the presence of an intact RF in the central canal enhances the spontaneous calcium activity of ventral CSF-cNs.

## Estimation of the elastic properties of the Reissner fiber from acute ablation

To gain a deeper understanding of the elastic properties of RF, we explored its response to acute photoablation. We performed RF photoablations via a UV-pulsed laser system in triple transgenic *Tg(sspo:sspo-GFP;pkd2l1:tagRFP;pkd2l1:GCaMP5G)* zebrafish larvae (N=74 fish). We then tracked the relaxation dynamics of the two cut ends of the fiber (*Figure 3*; see *Videos 7–9*; see also Materials and methods). We observed diverse kinetics and retraction patterns upon RF photoablation. The majority of fiber retractions (88% of 148 total retractions, *Figure 3A1 and A2*) demonstrated complete fiber retraction out of the 97 µm-wide field of view in less than 500 ms (initial retraction speed of on average ~328 µm/s), with 96% of

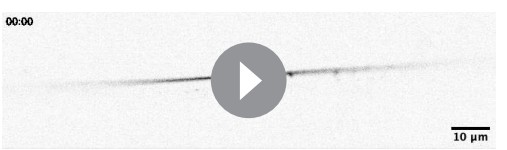

**Video 3.** Time series showing skeletal muscle contractions preventing the estimation of Reissner fiber motion in the sagittal plane of live *Tg(sspo:sspo-GFP)* unparalyzed transgenic larvae. Data was acquired at 10 Hz for 30 s. Rostral, left and dorsal, top. Video is replayed in real time (10 Hz). Scale bar represents 10 µm.
https://elifesciences.org/articles/86175/figures#video3

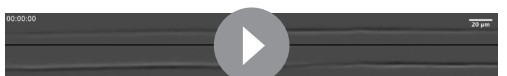

**Video 4.** Principal component analysis over all pixels of the time series reveal two main components of motion. The first principal component (top) corresponds to translation along the dorsoventral axis, and the second (bottom) corresponds to a small local rotation around the point in red in *Figure 1G2*. Data was acquired at 40 Hz for 25 s. Rostral, left and dorsal, top. Video is replayed in real time (40 Hz). Scale bar represents 20 µm.
https://elifesciences.org/articles/86175/figures#video4

**Video 5.** Spontaneous activity of cerebrospinal fluid-contacting neurons (CSF-cNs) in 3 dpf *Tg(sspo:sspo-GFP;pkd2l1:tagRFP; pkd2l1:GCaMP5G)* larvae before an acute ablation of the Reissner fiber. Calcium imaging was recorded between 3 and 4 Hz over 75 s. Rostral, left and dorsal, top. Video is replayed in real time (3.5 Hz). Scale bar represents 20 µm.

https://elifesciences.org/articles/86175/figures#video5

**Video 6.** Spontaneous activity of cerebrospinal fluid-contacting neurons (CSF-cNs) in 3 dpf *Tg(sspo:sspo-GFP;pkd2l1:tagRFP; pkd2l1:GCaMP5G)* larvae after an acute ablation of the Reissner fiber. Calcium imaging was recorded between 3 and 4 Hz over 75 s. Rostral, left and dorsal, top. Video is replayed in real time (3.5 Hz). Scale bar represents 20 µm.

https://elifesciences.org/articles/86175/figures#video6

fibers remaining straight as the cut ends of the fiber retracted to the rostral and caudal ends of the larvae. However, in a few cases (12% of 148 total retractions, *Figure 3B*) we observed the two cut ends of the fiber relax very slowly (initial retraction speed of on average ~50 µm/s), often with the cut ends remaining still in the field of view for over 20 s (*Figure 3C1–3C3*). In a third of these relaxed cases, we could observe that slow-retracting ablated fibers displayed at the tip snake-like deformations during their relaxation (*Figure 3B*). As an analogy with the dynamic model of DNA molecules (*Brochard-Wyart, 1995*), we refer to the deformed section of the fiber as the 'flower', which suggests the fiber was no longer under tension (*Figure 3B2*). In this analogy, the flower contrasts with the straight 'stem' section of the fiber that remained under tension (*Figure 3B3*).

To estimate the elastic properties of the fiber, in particular its mechanical diffusion coefficient, we developed a simple model for the ablation of an elastic fiber inspired by the dynamic model for DNA molecules (*Brochard-Wyart, 1995*). The RF can be seen as an elastic rod-like polymer in the central canal with a radius $r_f$ of ~100 nm (*Orts-Del'Immagine et al., 2020*). The relation between the pulling force $F_p$ acting on the fiber and its deformation can be described as:

$$F_p = \pi r_f{}^2 E \frac{\Delta L}{L} \tag{1}$$

with $E$ describing the elastic Young modulus and $\Delta L$ the elongation of the fiber stretched away from its original full length $L$ when it is under tension in vivo. When the fiber is cut, the deformation relaxes from the free end over a distance $x$ (e.g. 'the flower' while the rest of the fiber e.g. 'the stem' remains under tension). The size of the flower is deduced from a balance between the pulling force (*Equation 1*) and the friction force $F_v$ acting on the flower, which can be written as:

$$F_v = \frac{2\pi\eta}{\ln \frac{L}{r_f}} \times \frac{\partial x}{\partial t} \times \frac{\Delta L}{L} \sim \eta x \times \frac{\partial x}{\partial t} \times \frac{\Delta L}{L} \tag{2}$$

with $x$ being the size of the relaxed fiber (the flower), $\eta$ the CSF viscosity and $\frac{\partial x}{\partial t} \times \frac{\Delta L}{L}$ the retraction velocity that leads to friction. Using the slender-body approximation of the drag coefficient, the natural logarithm can be treated as a constant numerically. The force balanced, $F_p = F_v$ leads to a diffusion equation and therefore:

$$x^2 \sim Dt \tag{3}$$

where $D$ is given by $D = \frac{\pi r_f^2 E}{\eta} = \frac{r_f^2}{\tau}$ with the characteristic time $\tau$ for the fiber's mechanical relaxation $\tau^{-1} = \frac{\pi E}{\eta}$. Based on our model, the retraction distance upon ablation should increase as a function of $\sqrt{t}$. Because the deformation of the cut fiber is relaxed in the flower section, the retraction $X$ of the fiber given by the position of the free end is:

$$X = x \times \frac{\Delta L}{L} = \frac{2}{\sqrt{\pi}} \times \sqrt{Dt} \times \frac{\Delta L}{L} = \frac{2}{\sqrt{\pi}} \times r_f \sqrt{\frac{t}{\tau}} \times \frac{\Delta L}{L} \tag{4}$$

By plotting the retraction position as a function of $\sqrt{t}$ (*Figure 3C1*), we indeed found as expected from our simple model a linear relationship (*Figure 3C1*). The slope of this relationship defined by $\frac{2}{\sqrt{\pi}} \times \sqrt{D} \times \frac{\Delta L}{L}$ enabled us to group the fibers with fast retraction slope = 161 µm/√s and slow

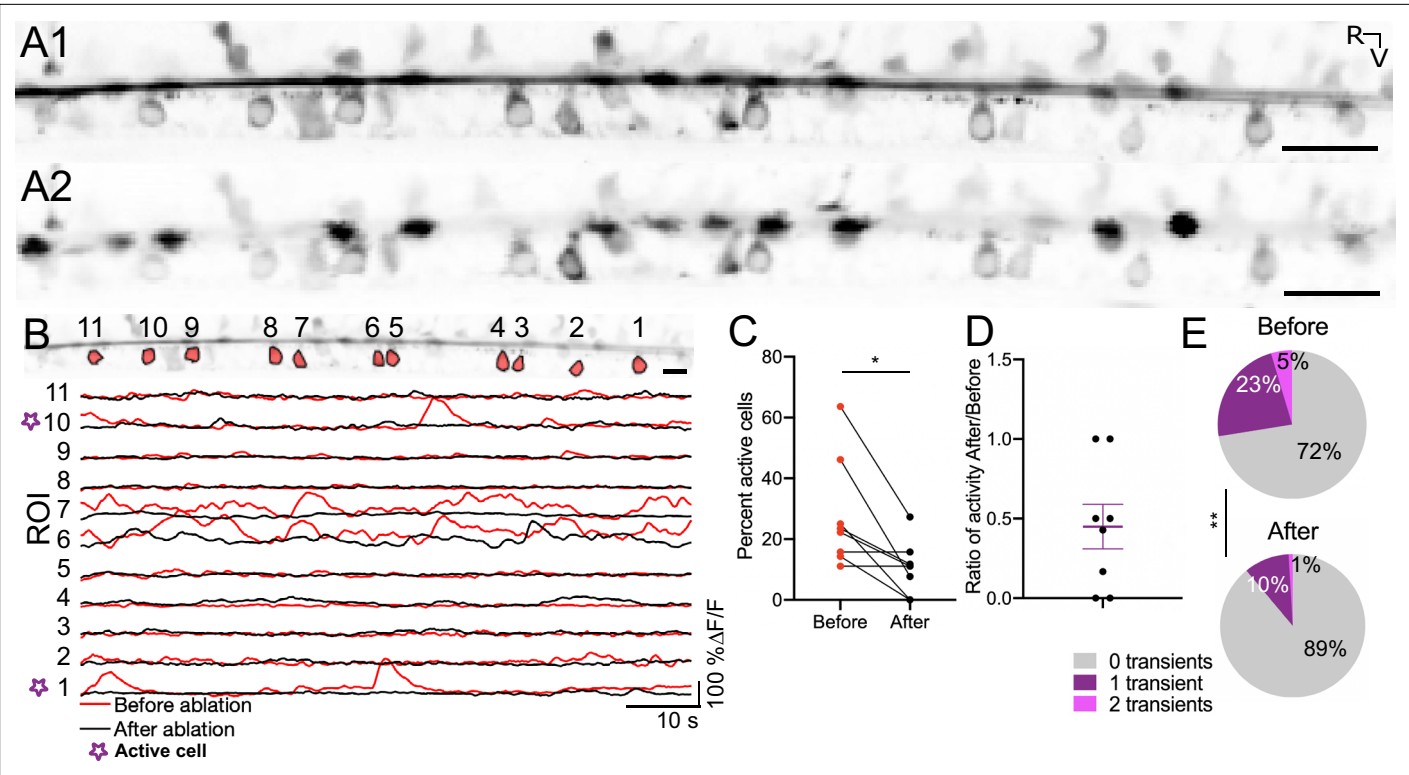

**Figure 2.** The Reissner fiber enhances spontaneous calcium activity in ventral CSF-cNs. (**A1**) Time-series standard deviation projection in the sagittal plane from two-photon laser scanning microscope showing the signal from the Reissner fiber and CSF-cNs in the central canal of 3 dpf *Tg(sspo:sspo-GFP;pkd2l1:tagRFP;pkd2l1:GCaMP5G)* zebrafish larva before RF photoablation. (**A2**) Time-series standard deviation projection in the sagittal plane from two-photon laser scanning microscope showing the signal from CSF-cNs after RF ablation performed by spiral scanning photoablation with an infrared pulsed laser tuned at 800 nm over 0.5 µm on RF (see Materials and methods). (**B**) ROI selection for ventral CSF-cNs to analyze activity before and after RF photoablation within the same cells (top). Example calcium activity traces normalized to baseline for each of the ROIs before (red) and after (black) RF photoablation over 75 s imaged at 3.45 Hz (see Materials and methods). (**C**) Percentage of active ventral CSF-cNs (active is defined as having at least 1 calcium transient during the recording) before and after RF photoablation (N=109 cells total from 8 fish from two independent clutches; mean percent active before ablation = 27.72% ± 6.34% versus mean percent active after ablation = 10.59% ± 3.1%; paired two-tailed t-test: p < 0.05). (**D**) The ratio of active ventral CSF-cNs after RF photoablation to those active before RF photoablation, illustrating on average, a fraction (on average ± SEM: 45% ± 14%) of active ventral CSF-cNs before photoablation remain active after RF photoablation. The purple lines on the graph represent the mean and the error bars indicate the SEM. (**E**) Pie charts illustrating the number of events per ventral CSF-cN before and after RF photoablation (mean number of events in active cells before RF photoablation = 0.94 events/min versus mean number of events in active cells after RF photoablation = 0.87 events/min; paired two-tailed t-test: p < 0.005). * p < 0.05, **p < 0.005. Scale bar is 20 µm (**A1, A2**), 10 µm (**B**).

retraction slope = 10 µm/√s (*Figure 3C1*). To assess whether the retraction kinematics of the fiber differ along the rostrocaudal axis, we compared the kinematics of retraction after acute RF photoablation performed at different sites (using the same terminology 'rostral', 'middle' and 'caudal', see *Figure 1A*). When ablations occurred in the rostral side, we only observed fast retraction kinetics (N=105 fiber retractions, *Figure 3D*). In contrast, ablations in the middle and caudal somites showed slower relaxation speeds (*Figure 3D*). The retraction speed for the rostral and caudal cut ends of a given fiber were highly correlated in individual larvae despite the diversity of retraction patterns observed overall across fish (y = 0.9x + 2; R² = 0.7; simple linear correlation: p < 10⁻⁴; *Figure 3D*), indicating that the retraction kinematics after ablation reveals the inherent physical properties of the fiber and the tension applied on it that differs as a function of the rostrocaudal position.

From the slope $\alpha = \frac{2}{\sqrt{\pi}} \times \sqrt{D} \times \frac{\Delta L}{L}$ of the retraction distance as a function of $\sqrt{t}$, we can extract from the information on the change of RF length ($\frac{\Delta L}{L}$) from the mechanical diffusion coefficient $D$. In ablation cases with retention of the fiber that remained in the field of view, we estimated $\Delta L$, the change of length of the fiber as it stretches away by the distance between the retention position of the cut end once immobile and the ablation locus. $\Delta L$ was negatively correlated with the position of

**Video 7.** Examples of retraction after laser ablation of the Reissner fiber used for *Figure 3A–C*. Position of RF was tracked using a spinning disk operating at 40 Hz for 25 s. Rostral, left and dorsal, top. Videos are replayed in real time (150 Hz). Scale bar represents 5 µm.
https://elifesciences.org/articles/86175/figures#video7

**Video 8.** Examples of retraction after laser ablation of the Reissner fiber used for *Figure 3A–C*. Position of RF was tracked using a spinning disk operating at 40 Hz for 25 s. Rostral, left and dorsal, top. Videos are replayed in real time (150 Hz). Scale bar represents 5 µm.
https://elifesciences.org/articles/86175/figures#video8

the ablation locus along the rostrocaudal axis (y = -0.9x + 41; $R^2$ = 0.2; simple linear regression: p < 0.06; *Figure 3E*). Accordingly, maximum retraction speeds were greater in the rostral somites of the larvae (on average ± standard deviation for all values provided hereafter: 448 µm/s ± 168 µm/s) than those in the middle or caudal somites (253 µm/s ± 166 µm/s and 211 µm/s ± 165 µm/s, respectively; Tukey's HSD Test for multiple comparisons; p < $10^{-4}$; *Figure 3F*).

In 13 cases of ablation on the caudal side, the cut ends of the fiber remained in the field of view during the 38 s-long recording and we could therefore measure $\Delta L$ as low as 10 µm. For rostral ablations, based on our field of view of 97 µm, we can estimate $\Delta L > 150$ µm (rostral ablation) on each side. These values indicate that for a fiber length of approximately 3 mm long, $\frac{\Delta L}{L} _{Rostral} \simeq \frac{1}{20}$ and $\frac{\Delta L}{L} _{Middle/Caudal} \simeq \frac{1}{300}$.

Using the slope $\alpha$, we grouped the fibers with fast retraction (slope = 161 µm/√s) versus the fibers with slow retraction (slope = 10 µm/√s; see *Figure 3D*). We found a remarkably-similar estimation of the mechanical diffusion coefficient for both groups with $D \simeq 5.6\ mm^2/s$ for fibers ablated on the rostral side and showing fast retraction: slope = 0.161 mm/√s, $\frac{\Delta L}{L} _{Rostral} \simeq \frac{1}{20}$; $D_{Rostral} \simeq 5.7 mm^2/s$; and for fibers ablated in the more caudal position and showing slow retraction: slope = 0.010 mm/√s; $\frac{\Delta L}{L} _{Middle/Caudal} \simeq \frac{1}{300}$; $D_{Middle/Caudal} \simeq 5.2 mm^2/s$. Consequently, the characteristic time for the fiber mechanical relaxation is $\tau = \frac{2}{\sqrt{\pi}} \times \frac{r_f^2}{D} \simeq 2\ ns$. The total retraction time is $T_{rup} = \frac{2}{\sqrt{\pi}} \times \frac{L^2}{D} = \tau$ and $\frac{L^2}{r_f^2} \simeq 200\ ms$, as observed experimentally.

The fit with experimental data shows that $D$ is uniform along the fiber, and faster retraction dynamics in the rostral rather than the caudal region demonstrates that the fiber tension increases along the rostrocaudal axis, from caudal to rostral. We can suggest two interpretations: the first one is the stretching of the fiber by frictional forces of the cerebrospinal fluid flowing from rostral to caudal with velocity $U$. At a distance $l$ from the caudal end, the hydrodynamic pulling force $\frac{2\pi \eta l U}{Ln \frac{L}{r_f}} \simeq \eta l U$, increasing from caudal ($l = 0$) to rostral ($l = L$), is balanced by the fiber tension $F_p(l) = \pi r_f^2 E \frac{\partial u}{\partial l}$, where $\frac{\partial u}{\partial l}$ is the fiber deformation. It leads to an increase in rostral deformation $\frac{\Delta L}{L} = \frac{\eta U L}{\pi r_f^2 E} \simeq \frac{U L}{D}$. The second contribution may be due to the stretching by the cilia of ependymal radial glial cells (the most numerous) exerting a force $f_p$ on the fiber in the flow direction. If $\nu$ is the linear density of the cilia-fiber links pulling with the force $f_p$, the resulting pulling force at a distance $l$ from the caudal end is $F_p(l) = \int \nu f_p \partial l = \nu f_p l$. The pulling force increases from the caudal to the rostral end, leading to a maximal rostral deformation $\frac{\Delta L}{L} = \frac{\nu f_p L}{\pi r_f^2 E}$. From the value of $D$, we can estimate the value of the fiber elastic modulus $E \sim \frac{Dn}{\pi r_f^2} \sim 10^6\ Pascal$. Estimation of the Young's modulus of biopolymers of the cytoskeleton gives E ranging from 1 to 4 GPa (*Brochard-Wyart et al., 2019*), a thousand times stiffer than the RF. Overall, the RF in larval zebrafish can be described as a soft elastic polymer that is maintained under tension in the CSF and exhibits a mechanical diffusion coefficient of about 5 mm²/s, a characteristic time in the order of 2 ns and an elastic modulus of $10^6$ Pascal – a value that would correspond to rubber rather than a rigid proteinaceous fiber.

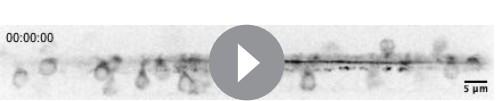

**Video 9.** Examples of retraction after laser ablation of the Reissner fiber used for *Figure 3A–C*. Position of RF was tracked using a spinning disk operating at 40 Hz for 25 s. Rostral, left and dorsal, top. Videos are replayed in real time (150 Hz). Scale bar represents 5 µm.
https://elifesciences.org/articles/86175/figures#video9

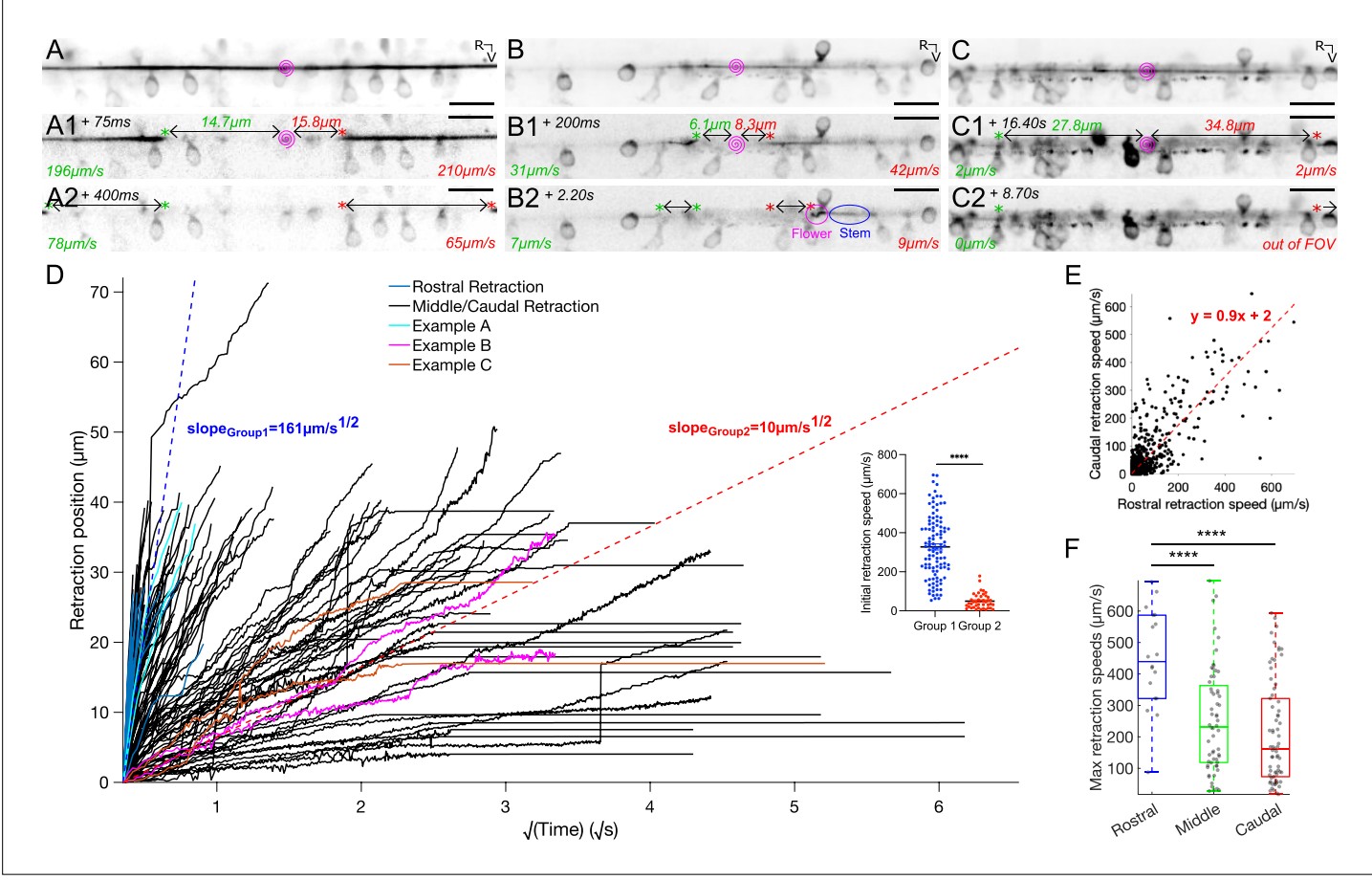

**Figure 3.** Reissner fiber photoablation reveals different modes of relaxation under tension. (**A**) Frame-by-frame instantaneous speed calculations for the rostral and caudal ends of the cut RF during retraction after photoablation by UV-pulsed laser in 3 dpf triple transgenic *Tg(sspo:sspo-GFP;pkd2l1:tagRFP;pkd2l1:GCaMP5G)* paralyzed larvae. Pink spiral indicates the site of RF ablation. Instantaneous speeds of the fiber for both rostral and caudal ends were calculated using the change in position of the fiber between two consecutive frames, divided by the exposure time (25 ms) and converted to μm/s. (**A1, A2**) specifically depict an example of a fiber in the 'rigid case', where the fiber retracts straight on both ends without distortion. (**B**) Instantaneous speed calculations for both rostral and caudal ends of a cut fiber in the 'relaxed case', where there is distortion of the fiber (**B1** left cut end, **B2** right cut end) due to no more tension on one or both ends of the fiber. By analogy with the dynamics of DNA molecules, one can refer to the completely relaxed portion of the fiber as the 'flower', while the remaining taut portion of the fiber would be the 'stem'. (**C**) Instantaneous speed calculations for both rostral and caudal ends of a cut fiber that remains stuck in the central canal, remaining in the field of view for about 25 s. (**D**) Retraction position plotted across the square root of time for all fish (N=74 fish from 7 independent clutches), color-coded to illustrate the examples in (**A–C**), and indicating if a flower was seen in the 38 s-long recordings along the rostrocaudal axis. $\sqrt{D} \times \frac{\Delta L}{L}$ is provided by the slopes of the two dotted lines (blue and red), indicating a fast retraction or a slow retraction of the rostral and caudal ends of the cut fiber in the central canal. Inset: initial retraction speed (the distance the cut fiber retracted in between 50 ms and 75 ms after photoablation, after UV laser artifacts) is larger in the fibers classified under the fast retraction group than those in the slow retraction group. (**E**) Rostral and caudal ends of the fiber in cases with and without flowers plotted against each other, demonstrating that rostral and caudal end dynamics generally tend to mirror each other (N=74 fish). (**F**) Maximum retraction speed for rostral and caudal ends of the cut RF classified using the schema in *Figure 1A*, grouping the ablations that occurred in the rostral somites (N=9 fish), middle somites (N=34 fish), and caudal somites (N=34 fish). Mean for ablation in rostral somites = 448 μm/s ± 168 μm/s versus in middle somites = 253 μm/s ± 166 μm/s versus in caudal somites = 211 μm/s ± 165 μm/s; Tukey's HSD Test for multiple comparisons: p < 10⁻⁴. ****p < 10⁻⁴. Scale bar is 10 μm (**A, B,C**).

## The Reissner fiber in the larva interacts with motile cilia along the central canal

The RF appears to directly interact with CSF-CN cilia in fixed tissues (*Orts-Del'Immagine et al., 2020*). Given that the RF displays diverse retraction patterns along the rostrocaudal axis after photoablation, we investigated whether the fiber may dynamically interact with cilia in the central canal in vivo (*Figure 4*). Ciliated cells in contact with the CSF include CSF-cNs, which project a short motile kinocilium (*Böhm et al., 2016*) and are known to be about a hundred at this stage (*Prendergast et al., 2023*), as well as ependymal radial glial cells, which project a longer monocilium (*Borovina et al.,*

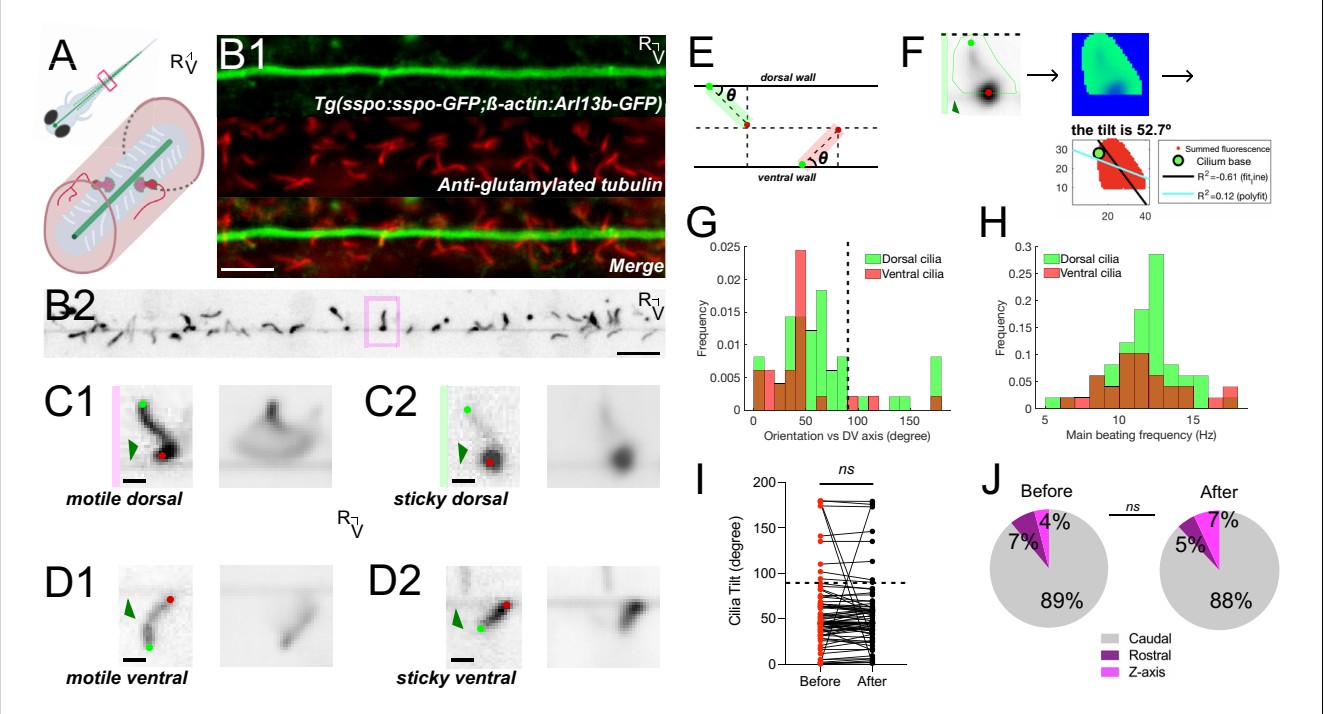

**Figure 4.** The Reissner fiber interacts with beating cilia protruding in the central canal. (**A**) Schematic illustrating motile cilia (white) lining the walls of the central canal beside CSF-cNs (red) with RF (green) bathing in the spinal cord of a zebrafish larva. (**B1**) Z-stack projection in the sagittal plane showing the immunostaining against GFP and glutamylated tubulin in a *Tg(sspo:sspo-GFP;-ß-actin:Arl13b-GFP)* larva fixed at 3 dpf revealing respectively RF and cilia labeled by the ß-actin promoter (both in green here) and motile cilia (red). Rostral to the left, ventral displayed on the bottom. (**B2**) Single optical section in the sagittal plane showing RF surrounded by cilia protruding in the central canal in 3 dpf *Tg(sspo:sspo-GFP;ß-actin:Arl13b-GFP)* paralyzed larva out of the streaming video acquired at 40 Hz using spinning disk confocal microscopy setup equipped with a 40X objective. (**C1**) Right: example single optical section of a dorsal motile cilium from the highlighted box in (**B1**). Dark green arrow indicates RF. Light green and red dots indicate the base and tip of the cilium, respectively. Left: average projection over 25 s of the dorsal cilium brushing against RF. (**C2**) Right: example single optical section of a dorsal motile cilium whose tip tends to stick to RF. Left: average projection over 25 s of the dorsal cilium. (**D1**) Right: example single optical section of a ventral motile cilium whose tip tends to brush against RF. Left: average projection over 25 s of the ventral cilium. (**D2**) Right: example single optical section of a ventral motile cilium whose tip tends to stick to RF. Left: average projection over 25 s of the ventral cilium. (**E**) Schematic illustrating the geometric calculation of ciliary orientation versus the dorsoventral axis. (**F**) Representative analysis process for the dorsal cilium in (**C2**). A mask was first drawn over a single cilium to calculate the frequency and orientation in a specific region of interest (left). A temporal mean of that region (middle) was then used to calculate the orientation of the cilium in respect to the dorsoventral axis (right; see also Materials and methods). (**G1**) Distribution of the orientation in respect to the dorsoventral axis of dorsal mean ± standard deviation provided hereafter: (64.8° ± 44.4°; N=49 cilia across 8 fish) and ventral (43.9° ± 35.8°; N=27 cilia across 9 fish) motile cilia. (**G2**) Distribution of main ciliary beating frequency for dorsal (mean ± standard deviation provided hereafter: 11.8 Hz ± 2.7 Hz; N=49 cilia) and ventral (11.6 Hz ± 2.8 Hz; N=27 cilia) motile cilia. (**H**) The orientation of motile cilia in the central canal was not significantly different after RF photoablation (mean ± standard deviation provided hereafter: 55.1° ± 39.3°) from that before RF photoablation (57.4° ± 42.5°), illustrating that cilia orientation is, on average, not significantly affected by RF photoablation (N=76 cilia from 9 fish; paired two-tailed t-test: p > 0.3). (**J**) Overall rostrocaudal polarity of motile cilia remained similar before and after RF photoablation, with about 90% of motile cilia polarized toward the caudal end of the fish, and the remaining 10% either polarized toward the dorsal end of the fish or beating in the Z axis (N=76 cilia from 9 fish; paired two-tailed t-test: p > 0.3). ns = not significant. Scale bar is 10 μm (**B1, B2**) and 2 μm (**C1, C2, D1, D2**).

The online version of this article includes the following figure supplement(s) for figure 4:

**Figure supplement 1.** Changes in the main ciliary beating frequency varied from fish to fish in response to RF photoablation.

**Figure supplement 2.** Motile cilia position profile differs with Reissner fiber dorsoventral oscillation position.

**Figure supplement 3.** Example cilia analysis process.

*2010*; *Becker and Becker, 2015*) and are estimated via electron mictroscopy (*Orts-Del'Immagine et al., 2020*) and the transgenic *Tg(ß-actin:Arl13b-GFP)* line *Sternberg et al., 2018* to have a density of about 60 cilia per 10 μm along the 3.5 mm-long central canal, encompassing at least ~20,000 cilia at this stage. We asked whether the latter may interact with RF to elicit the apparent friction and build a heterogenous tension along the fiber.

**Video 10.** Motile cilia beating and interacting with the Reissner fiber in the *Tg(sspo:sspo-GFP;β-actin:Arl13b-GFP)* larvae. Position of the cilia and RF were tracked using a spinning disk operating at 40 Hz for 25 s. Note the star symbol indicates an example motile cilium brushing against RF in sweeping motions. Rostral, left and dorsal, top. Video is replayed in real time (40 Hz). Scale bar represents 5 μm.
https://elifesciences.org/articles/86175/figures#video10

**Video 11.** Motile cilia beating after laser-mediated ablation of the Reissner fiber in the *Tg(sspo:sspo-GFP;β-actin:Arl13b-GFP)* larvae. Orientation of the cilia relative to the horizontal were tracked from data acquired on a spinning disk operating at 40 Hz for 25 *s*. Rostral, left and dorsal, top. Video is replayed in real time (40 Hz). Scale bar represents 5 μm.
https://elifesciences.org/articles/86175/figures#video11

We investigated whether we could find evidence for an interaction between RF and motile cilia using high-speed imaging in double transgenic *Tg(sspo:sspo-GFP;β-actin:Arl13b-GFP)* larvae, in which many cilia in the central canal are labeled with GFP (*Figure 4A and B1*; see also Methods).

Immunohistochemistry staining against glutamylated tubulin and GFP showed that motile cilia are densely packed along the dorsal and ventral walls of the central canal and are oriented with a caudal tilt (*Figure 4B1*). In contrast to the embryo, in which only ventral cilia are motile and tilted towards the caudal end (*Thouvenin et al., 2020*), we observed that motile cilia in 3 dpf larvae are located both on the ventral and dorsal walls of the central canal at larval stage (*Figure 4B2*; see also *Video 10*). When we took a closer look at isolated motile cilia labeled in 3 dpf *Tg(β-actin:Arl13b-GFP)* larvae and beating in the sagittal plane of imaging (see Materials and methods), we observed that numerous motile cilia brushed against RF in sweeping motions (*Figure 4B2, C1 and D1*; see also star symbol in *Video 10*), while others appear to almost stick to RF with a glob of fluorescent material at their tip (*Figure 4C2 and D2*; see also *Video 10*). We quantified the orientation of cilia in an angle relative to the horizontal (*Figure 4E and F*), and observed an overall tilt towards the caudal end both for dorsal inserted cilia (mean ± standard deviation provided hereafter: 64.8° ± 44.4°; N=49 cilia across 8 fish) and ventral inserted cilia (43.9° ± 35.8°; N=27 cilia across 9 fish; *Figure 4G*). We estimated the main ciliary beating frequency to be on average 11.2 Hz ± 2.4 Hz overall (*Figure 4H*; dorsal: 11.8 Hz ± 2.7 Hz; ventral: 11.6 Hz ± 2.8 Hz). However, we were limited in our acquisition frequency (40 Hz) in the setup that was suitable for ablation with a pulsed laser, thus the range of main beating frequencies up to only 20 Hz for both dorsal and ventral motile cilia may not reflect the actual beating frequency of these cilia (*Figure 4H*).

To investigate the impact of the RF on the beating frequency of motile cilia in the central canal, we performed acute focal ablations of the RF (see Methods). We quantified the orientation and main beating frequency of the cilia before and after RF photoablation (*Figure 4H*; see also *Figure 4—figure supplement 1A* and *Video 11*). The tilt of cilia was not significantly different after RF photoablation (mean ± standard deviation provided hereafter: 55.1° ± 39.3°) from that before RF photoablation (57.4° ± 42.5°), illustrating that cilia orientation was, on average, not significantly affected by RF photoablation (*Figure 4J*; N=76 cilia from 9 fish; paired two-tailed t-test: p > 0.3). In contrast, the impact of acute RF photoablation on ciliary beating frequency showed different responses among individuals (*Figure 4—figure supplement 1B1–B3*). The fact that motile cilia densely surround the fiber along the walls of the central canal and brush along the fiber with a caudal tilt suggests that cilia-fiber interactions may generate friction promoting the oscillatory deflections of RF observed in vivo (*Figure 1*) and the deformations of the 'flower' ends of the cut RF during RF relaxation after photoablation (*Figure 3B, B1 and B2*).

To investigate this further, we performed independent k-means clustering to relate the beating orientation versus the dorsoventral axis of individual cilia along the rostrocaudal axis with the dorsoventral oscillatory motion of RF (*Figure 4—figure supplement 2*). While dorsal and ventral motile cilia position profiles differ with RF dorsoventral oscillation position, a minority of cilia (N=7/15 dorsal and N=4/9 ventral) show significant differences in their beating orientation depending on the up or down position of RF (*Figure 4—figure supplement 2B1, B2*; Fisher's exact test: p < 0.05 and p < 0.01 for dorsal and ventral example cilia, respectively). Of these cilia, both dorsal and ventral cilia were closer to the vertical orientation when the fiber was down compared to when it is up (mean ± standard

deviation from hereafter: 60° ± 10° when RF down versus 48° ± 16° when RF up for dorsal cilia, and 55° ± 23° when RF down versus 33° ± 16° when RF up for ventral cilia; *Figure 4—figure supplement 2C*). Our current dataset should be extended in 3D to find out whether beating cilia collectively exert push or pull interactions onto the fiber, in order to explain the heterogenous and graded tension we observed on RF, with larger tension on the rostral side.

## Discussion

Our work reveals that the Reissner fiber (RF) is a dynamic structure under tension in vivo in the central canal with elastic properties and spontaneous oscillatory activity. Our mechanical model reveals that the Reissner fiber in larval zebrafish can be described as a soft elastic polymer that is maintained under tension in the CSF and exhibits a mechanical diffusion coefficient of 5 mm²/s, a characteristic time in the order of 2 ns and an elastic modulus of $10^6$ Pascal, which would fit more with the fiber acting as a low density gel rather than a rigid proteinaceous fiber. At baseline, in paralyzed animals with a straight body axis, we found evidence that the Reissner fiber interacts with the numerous long, beating monocilia of ependymal radial glial cells (ERGs), which we estimated to have a density of at least 20,000 cilia at this stage (*Orts-Del'Immagine et al., 2020*; *Sternberg et al., 2018*) as well as with some ciliated sensory neurons (CSF-cNs), known to be only about a hundred at this stage (*Prendergast et al., 2023*), whose activity decreases upon photoablation of the fiber.

### The Reissner fiber under tension oscillates along the dorsoventral axis in the central canal

By investigating the dynamical properties of the Reissner fiber in the central canal, we observed that the fiber is under tension in vivo, confirming previous observations of RF being rectilinear (*Troutwine et al., 2020*), which contrasts with observations in the tissue after fixation, in which it curls and bends across the rostrocaudal axis (*Orts-Del'Immagine et al., 2020*). Our observations demonstrate that RF can be modeled as a taut polymer under a heterogeneous tension in vivo. In addition to the spontaneous slow translation of RF previously observed along the rostrocaudal axis with material continually added and retracted from its surface (*Troutwine et al., 2020*), our observations further reveal spontaneous oscillatory activity of the fiber over the dorsoventral axis. We observe large spontaneous oscillatory activity of RF in the middle portion of the fish, away from the attachment points in the rostral end (SCO) and caudal end (ampulla caudalis) (*Meiniel and Meiniel, 2007*; *Gobron et al., 2000*), akin to oscillatory behaviors of a plucked guitar string.

To capture enough photons from the GFP-tagged fiber in our recordings, we sampled displacements of the fiber along the dorsoventral axis at 40 Hz at most. However, we have indications that the dorsoventral oscillations of the fiber probably occur at higher frequency than the 20 Hz we could observe in these conditions. Previously, the coordinated beating of motile cilia in the central canal has been reported up to 45 Hz, generating CSF flow estimated at a velocity of 10 µm/s in the embryo (*Thouvenin et al., 2020*). Such beating of cilia may influence locally how the Reissner fiber oscillates across the dorsoventral axis. Furthermore, interactions between motile cilia and the fiber may contribute to maintaining the fiber under tension, with graded tension on the rostral end. Further investigations of the 3D interactions between the Reissner fiber and motile cilia in the CSF will be necessary to better understand the role of these interactions for the dynamic and physical properties of the Reissner fiber.

### Elastic properties of the Reissner fiber subject to a heterogeneous tension decreasing along the rostrocaudal axis

Acute focal ablation of RF allowed for the estimation of its elastic properties in the central canal. Upon ablation, the fiber retracted both further and faster when the ablation occurred on the rostral side of the fish compared to when the fiber was cut in the middle or caudal portions. A simple model of RF represented as an elastic polymer under tension enabled us to estimate RF mechanical diffusion coefficient $D \sim 5\ mm^2/s$ and RF characteristic mechanical time $\tau \sim 2\ ns$. The full retraction time after RF rupture is proportional to $\tau$ and to a huge factor $(\frac{L}{r_f})^2$, with $r_f$ of $\sim 100\ nm$ and $L = 1\ mm$. The value of the stress diffusion coefficient $D$ being highly conserved in the rostral and middle/caudal portions of the fiber suggests that the diameter of the fiber is constant along the rostrocaudal axis.

In contrast, the faster retraction speeds observed after fiber ablation on the rostral side reveals that the pulling force on the fiber may not be uniform along the rostrocaudal axis. A higher tension in the rostral portion may be due to external factors increasing tension on the rostral side, such as the CSF flow going in the rostrocaudal direction (*Tumani et al., 2018*; *Zhang et al., 2018*) and/or the friction associated with the numerous interactions occurring between the fiber and the beating cilia along the central canal.

## Interactions between the Reissner fiber and ciliated cells in the central canal

We observed in streaming acquisitions of RF and cilia labeled in GFP that some motile cilia brush and interact with the fiber. Accordingly, we measured that motile cilia from different fish have different beating frequencies in response to RF photoablation. Altogether, our observations indicate that RF interacts with motile cilia, a mechanism that can lead to a reduction of ciliary beating as well as to friction between RF and the cilia. The impact of RF ablation on cilia beating frequency is still elusive, which is consistent with our previous observations that the absence of a fiber did not lead to a massive change in particle velocity profile in the embryo (*Cantaut-Belarif et al., 2018*). However, the fiber could nonetheless slightly alter the CSF flow profile in a transverse section by adding a point of null flow at the center of the central canal. Adding a point of null velocity can increase the velocity gradient in the CSF, and thereby increase the phenomenon of shearing at the level of the CSF-cN apical extension - in line with their functional coupling as reported was critical to mediate mechanoreception in vivo (*Orts-Del'Immagine et al., 2020*).

Since the Reissner fiber radius is ~100 nm (*Orts-Del'Immagine et al., 2020*), our resolution to identify the center of the fiber with classical fluorescent microscopy is certainly limited. Similarly, the precise mechanism of interaction between RF and the ciliated neurons (CSF-cNs) cannot be finely resolved to decipher between direct contacts and a change of CSF flow at the level of the apical extension of the CSF-cNs. Nonetheless, we showed here that acute ablation of the Reissner fiber reduced the spontaneous calcium activity of CSF-cNs, indicating that a functional coupling may exist as well when the fiber spontaneously oscillates. Similarly, our analysis of ciliary beating was limited by the temporal resolution of our recordings on the confocal microscope equipped with a pulsed laser for ablation. A finer analysis of the spatiotemporal kinematics of the ciliary beating should resolve whether the Reissner fiber interacts preferentially with cilia beating at a given frequency, a given orientation or their position of insertion in the central canal.

In this study, we made a new step in understanding the axial sensory system composed of the Reissner fiber and the CSF-cNs. We demonstrated that the Reissner fiber at rest, when the body is straight, spontaneously oscillates and enhances spontaneous calcium activity in ventral CSF-cNs. The fiber's vertical movement along the dorsoventral axis could lead to contact with the CSF-cN apical extension to enhance the spontaneous calcium activity of the ventrally located sensory neurons we measured here. Interactions with dorsally located sensory neurons may occur as well due to displacement in the horizontal plane, but were more difficult to image and report on in our conditions. The axial sensory system could form around the oscillating fiber and lead to the observed noise in the neuronal activity in resting baseline position (*Sternberg et al., 2018*; *Prendergast et al., 2023*).

In zebrafish with genetic mutations, silencing or acute manipulations affecting CSF-cN activity or the maintenance of the Reissner fiber, sensorimotor deficits included slowing down of locomotion (a reduction of locomotor frequency and amplitude, *Böhm et al., 2016*; *Wu et al., 2021*), postural defects during fast locomotion (rolling, *Hubbard et al., 2016*; *Wu et al., 2021*) as well as morphological defects of the body axis in the embryo (*Cantaut-Belarif et al., 2018*) and the spine in juvenile/adult fish (*Sternberg et al., 2018*; *Troutwine et al., 2020*; *Rose et al., 2020*). Multiple studies in mice indicate that CSF-cN-dependent functions for sensorimotor integration during locomotion observed in zebrafish are conserved in mammals (*Gerstmann et al., 2022*; *Nakamura et al., 2023*). The SCOspondin gene SSPOP is a pseudogene in humans (https://www-ncbi-nlm-nih-gov/gene/23145), supporting the hypothesis that the Reissner fiber is absent in humans and great apes. However, evidence of urotensin signaling being relevant in human patients with adolescent idiopathic scoliosis (*Dai et al., 2021*; *Xie et al., 2023*) raises the question of whether the urotensin signaling pathway involves CSF-cNs in order to contribute to morphogenesis and body axis straightening. Future studies

will address whether acute ablation of the Reissner fiber can lead to similar deficits in sensorimotor integration for locomotion, posture and morphogenesis.

# Materials and methods

**Key resources table**

| Reagent type (species) or resource | Designation | Source or reference | Identifiers | Additional information |
|---|---|---|---|---|
| Gene (Zebrafish) | *Tg(sspo: sspo-GFP)* | *ut24Tg*; **Troutwine et al., 2020** | | |
| Gene (Zebrafish) | *Tg(pkd2l1: tagRFP)* | *icm17Tg*; **Böhm et al., 2016** | | |
| Gene (Zebrafish) | *Tg(pkd2l1: GCaMP5G)* | *icm07Tg*; **Böhm et al., 2016** | | |
| Gene (Zebrafish) | *Tg(β-actin: Arl13b-GFP)* | *hsc5Tg*; **Borovina et al., 2010** | ZFIN: ZDB-ALT-100721–1 | Referred to as *Tg(β-actin: Arl13b-GFP)* in this paper |
| Antibody | Anti-GFP (Chicken polyclonal) | Abcam | Cat# ab13970; RRID: AB-300798 | IHC(1:400) |
| Antibody | Anti-Polyglutamylation Modification (GT335) (Mouse monoclonal) | Adipogen | Cat# AG-20B-0020-C100; RRID: AB-2490210 | IHC(1:400) |
| Antibody | Alexa Fluor-488 (Goat anti-chicken polyclonal) | Thermo Fisher Scientific | Cat# A-11039; RRID: AB-2534096 | IHC(1:500) |
| Antibody | Alexa Fluor-568 (Goat anti-mouse polyclonal) | Thermo Fisher Scientific | Cat# A-11004; RRID: AB-2534072 | IHC(1:500) |
| Chemical compound, drug | α-Bungarotoxin | TOCRIS | Cat# 2133 | |
| Chemical compound, drug | Tricaine (MS 222) | Sigma-Aldrich | Cat# E10521 | |
| Chemical compound, drug | Paraformaldehyde solution (PFA) | Delta Microscopy | Cat# 15714 | |
| Chemical compound, drug | Phosphate buffered saline | Thermo Fisher Scientific | Cat# BR0014G | |
| Chemical compound, drug | Triton X-100 | Merck | Cat# 1086031000 | |
| Chemical compound, drug | Bovine Serum Albumin (BSA) | Sigma | Cat# A7030 | |
| Chemical compound, drug | Normal Goat Serum (NGS) | Sigma | Cat# NS02L | |
| Chemical compound, drug | Dimethyl sulfoxide (DMSO) | Sigma | Cat# D8418 | |
| Chemical compound, drug | Vectashield | Vector Laboratories | Cat# H-1000–10 | |
| Chemical compound, drug | Glycerol | VWR | Cat# 24387.292 | |
| Chemical compound, drug | Acetone | VWR | Cat# 20066.296 | |
| Software, algorithm | ImageJ/Fiji | **Schindelin et al., 2012** | https://imagej.net/ | |
| Software, algorithm | MATLAB | The MathWorks Inc. | https://www.mathworks.com/ | |
| Software, algorithm | GraphPad Prism | GraphPad | https://www.graphpad.com/ | |

*Continued on next page*

*Continued*

| Reagent type (species) or resource | Designation | Source or reference | Identifiers | Additional information |
|---|---|---|---|---|
| Other | DAPI stain | Molecular Probes | Cat# D1306; RRID: AB-2629482 | IHC(1:1000) |
| Other | iLas Pulse | Gataca Systems | gataca-systems.com | UV Laser Ablation |

### Materials availability statement

Further information and requests for resources and reagents should be directed to and will be fulfilled by the corresponding author Claire Wyart (claire.wyart@icm-institute.org). Note that this study did not generate new unique reagents. All data and codes generated and analyzed during this study can be found on Dryad (https://doi.org/10.5061/dryad.573n5tbc2).

### Experimental model and subject details

All imaging procedures were performed on 3 days post fertilization (dpf) zebrafish larvae in accordance with the European Communities Council Directive (2010/63/EU) and French law (87/848) and approved by the Paris Brain Institute (Institut du Cerveau, ICM). All experiments were performed on *Danio rerio* embryos of AB, Tüpfel long fin (TL) and nacre background. Animals were raised at 28.5 °C under a 14/10 light / dark cycle until the start of the experiment. All analyses were performed on animals that were in good health during the experiments.

### Method details

All imaging experiments were done on 3 dpf *Tg(sspo:sspo-GFP)*, *Tg(sspo:sspo-GFP;pkd2l1:tagRFP;pkd2l1:GCaMP5G)*, and *Tg(sspo:sspo-GFP;β-actin:Arl13b-GFP)* zebrafish larvae. Their respective siblings were used as controls (i.e., wild-type and heterozygous mutants from the same clutch). For all experiments, larvae were laterally mounted in glass-bottom dishes (MatTek, Ashland, Massachusetts, USA), filled with 1.5% low-melting point agarose. Unless otherwise noted, larvae were paralyzed by injecting 1–2 nL of 500 mM alpha-bungarotoxin (TOCRIS) in the caudal muscles of the trunk via glass micropipette held by a micromanipulator (Märzhäuser Wetzlar MM-33), using a pneumatic Pico-pump (World Precision Instruments PV-820).

#### Live imaging of the Reissner fiber

Three dpf larvae from *Tg(sspo:sspo-GFP)* incrosses were laterally mounted and paralyzed. An inverted spinning disk confocal microscope (Leica/Andor) equipped with a 40X water immersion objective (N.A.=0.8) was used to acquire images at 40 Hz for 25 s. Fish were laterally sampled along the rostro-caudal axis from somites 1–30 to gain a holistic understanding of the Reissner fiber's movements over the dorsoventral axis. The same procedure was performed with 3 dpf larvae from *Tg(sspo:sspo-GFP;β-actin:Arl13b-GFP)* double-transgenic larvae to image the interaction between the Reissner fiber with motile cilia in the central canal.

#### Tracking of the Reissner fiber

To estimate the fiber curve $x \mapsto y(x)$ on each image, we implemented the following algorithm.

Step 1: First, we estimated the background level by analyzing the histogram of the whole sequence. Calling $I_0$ the intensity associated to the maximum value in the histogram (an approximation of the average background level), and $I_1$ the average of all intensities smaller than $I_0$ (so that $D = I_0 - I_1$ is an approximation of the average absolute deviation of the background), we subtracted the value $I_0 + D$ to the whole sequence and set negative values to 0. This first step removes a lot of background noise from the sequence, without affecting much of the fiber signal. Then, we applied the following processing to each image of the sequence.

Step 2: We convoluted the image with a two-dimensional (non-isotropic) Gaussian filter to increase the signal-to-noise ratio. Since the fiber local orientation is always close to horizontal and has slow variations, we used a rather large horizontal width $\sigma_x = \frac{15}{3\sqrt{2}} \sim 3.5 \ pixels$ and a moderate vertical width $\sigma_y = \frac{8}{3\sqrt{2}} \sim 1.9 \ pixels$.

Step 3: For each column x, we computed the integer position $y_0$ of the pixel with maximal intensity $I(x, y_0)$, and used the intensities of the pixels above and underneath to get a sub-pixellic refinement $y_{raw}(x)$ of $y_0$ using a second order polynomial fit: $y_{raw}(x) = \frac{y_0(x) + \min(1, \max(-1, (I(x, y_0+1) - I(x, y_0-1)))}{4I(x, y_0) - 2I(x, y_0-1) - 2I(x, y_0+1)}$.

Step 4: At this stage, we also estimated the local fiber width as the diameter of the (one-dimensional) region obtained by thresholding the intensity at half the maximum value encountered on the current column. This value was used to classify columns as valid (domain $D$) when the width was less than 10 *pixels*, and invalid otherwise. This step is useful to remove uncertain position estimates due to firing neighboring cells: if both the fiber and a firing cell are visible (with similar intensity levels) on a given column $x$, the estimated width is too large and $x$ will not belong to $D$.

Step 5: We then initialized $y(x)$ to $y_{raw}(x)$ for all $x$ and iterated until convergence a 2-step procedure: $y(x) \leftarrow y_{raw}(x)$ for all $x$ in $D$ (enforce valid values for $y$) $y(x) \leftarrow \frac{(y*G)(x)}{(1*G)(x)}$ (smooth $y$ using a convolution with a Gaussian kernel $G$). It is not difficult to prove that this iterative algorithm converges (independently of the initialization of $y$), and the convergence is quite fast. In practice, we use a Gaussian kernel $G$ with $\sigma \sim 5.8$, and 100 iterations were more than enough. Step 5 has two effects: it smooths the function $y_{raw}$ and extrapolates it outside the domain $D$.

Step 6: In the (rare, but possible) case when there is a firing cell and an non-visible (or too faint) fiber on a given column $x$, Step 4 may wrongly classify $x$ as a valid column. We can detect such a column $x$ using the fact that the wrong estimate $y_{raw}(x)$ will generally be inconsistent with the expected fiber smoothness, and thus depart significantly from its corresponding smooth estimate $y(x)$. In practice, we removed from set $D$ all columns $x$ for which $|y(x) - y_{raw}(x)|$ was larger than 1 pixel. With this restricted set $D$, we repeated the previous algorithm of Step 5 (iteration until convergence of the 2-steps procedure) to obtain the final estimate of the fiber curve $x \mapsto y(x)$.

The algorithm is made available as a MATLAB function. To prepare the acquired raw data to run the script, videos were cropped to the dimensions of roughly 15 µm x 100 µm, to have the Reissner fiber in the middle of the field of view with its surrounding background. Videos were discretized into 2 µm bins along the rostrocaudal axis prior to running the script. When calculating dorsoventral displacement from the mean position of the fiber, videos were pieced apart and cropped to the dimensions of roughly 15 µm x 20 µm over 25 s five times, to encompass the whole recording cropped to 15 µm x 100 µm. The script was then run and dorsoventral displacement was saved to different variables and concatenated in the end to see the holistic dorsoventral displacement from the mean of the fiber for each fish.

## Two-photon ablation of the Reissner fiber and calcium imaging of ciliated sensory neurons

3 dpf larvae from *Tg(sspo:sspo-GFP;pkd2l1:tagRFP; pkd2l1:GCaMP5G)* incrosses were laterally mounted and paralyzed. Two-photon spiral scanning ablations at 800 *nm* were performed with a two-photon laser scanning microscope (2p-vivo, Intelligent Imaging Innovations, Inc, Denver, Colorado, USA) equipped with a 20X objective (N.A.=1.0). A spiral scanning ablation at 800 nm over 0.5 µm was programmed within the two-photon photomanipulation settings before capturing the acquisition. Fish were laterally sampled using a 920 nm IR laser at an imaging frequency between 3–4 Hz over 75 s along the rostrocaudal axis from somites 1–30 to gain a holistic understanding of the impact of RF ablation on CSF-cN activity before and after ablation along the central canal. Calcium transients of the CSF-cNs were analyzed by improving a previously developed MATLAB function in order to reduce variability in the detection of the baseline in larvae compared to embryos (*Sternberg et al., 2018*). Videos were first renamed and renumbered in order to perform the analyses blindly. Videos were then registered to correct for any motion artifacts, and then run through the script to determine CSF-cN ROIs and their respective baselines for their calcium transient traces. CSF-cNs were only designated as active if there was a completed calcium transient greater than 3 standard deviations above the baseline over the 75 s recording.

## Ablation of the Reissner fiber with a UV pulsed laser for kinematic analysis of retraction

Three dpf larvae from *Tg(sspo:sspo-GFP;pkd2l1:tagRFP; pkd2l1:GCaMP5G)* incrosses were laterally mounted and paralyzed. An inverted CSU-X1 microscope (Yokogawa, Japan) equipped with a 40X oil immersion objective (N.A.=1.3) was used to acquire images at 40 Hz for 25 s. A live FRAP

photoablation (iLas Pulse, Gataca Systems, Massy, France) of RF with an 8 nm pulse diameter and 20 ms duration was manually triggered in the middle of the acquisition. Prior to ablation, approximately 20 µm Z stacks (step size = 0.3 µm) were taken of the field of view at 10 Hz to gain a sense of the surroundings of the Reissner fiber at that position in the central canal. Fish were laterally sampled along the rostrocaudal axis from somites 1–30 to gain a holistic understanding of the Reissner fiber's behavior upon ablation along the central canal. Subsequent analysis consisted of measuring the frame-by-frame relaxation of each end of the cut fiber, which was performed by a MATLAB function. Prior to running the script, videos were first rotated to have the Reissner fiber aligned horizontally at 0°, and then cropped to only include frames just before the UV ablation and until the fiber is out of the field of view.

## Ablation of the Reissner fiber with a UV-pulsed laser with motile cilia imaging and analysis

A similar protocol as above was employed to ablate the Reissner fiber while observing the dynamics of the motile cilia. 3 dpf larvae from *Tg(sspo:sspo-GFP;β-actin:Arl13b-GFP)* double-transgenic larvae were laterally mounted and paralyzed. An inverted CSU-X1 microscope (Yokogawa, Japan) equipped with a 40X oil immersion objective (N.A.=1.3) was used to acquire images for 25 s. Previous work at an imaging frequency of 100 Hz has shown main beating frequencies of motile cilia in the central canal ranging up to 45 Hz in 30 hours post fertilization zebrafish embryos (*Thouvenin et al., 2020*; *Thouvenin et al., 2021*); however, in order to see the fluorescent signal of RF at the larval stage, we performed these experiments at an imaging frequency of 40 Hz. A live FRAP photoablation (iLas Pulse, Gataca Systems, Massy, France) of RF with an 8 nm pulse diameter and 20 *ms* duration was manually triggered in the middle of the acquisition. To analyze the resulting images, we implemented the following process:

Step 1: First, we selected individual dorsal and ventral motile cilia in sparse areas of each video across 9 fish. Cilia that were chosen fulfilled the following criteria: (1) the individual cilium could be seen in focus (even if dimly fluorescent) during the duration of the video, and (2) the cilium was in a sparse enough environment to analyze only its behavior. Cilia that were overlapping were therefore excluded from our analyses, since we could not distinguish distinct features from one cilium at a time if they were in this configuration. On average, approximately nine individual cilia were selected in each recording and cropped in an approximately 10 µm x 10 µm square, to only get one specific cilium in the field of view (*Figure 4—figure supplement 3*).

Step 2: If after cropping another cilium or the Reissner fiber was still in the field of view, we would mask parts of the video that had additional information via a MATLAB script. We drew a circle around the cilium of interest and masked the area outside the region of interest, to ultimately only run the analysis on the specific cilium of interest. The background of the video was thresholded if the cilium was dimly fluorescent to increase signal and analysis accuracy.

Step 3: To calculate the ciliary beating frequency, the raw fluorescent trace of the signal inside a region of interest covering a motile section of the cilia was plotted. The main ciliary beating frequency in Hz was estimated by counting the number of oscillations in the fluorescent signal per second.

Step 4: The cilium's orientation in respect to the dorsoventral axis was then calculated. Values from –90° to +90° relative to the horizontal with 0° toward the caudal end correspond to a caudal tilt of ventral cilia, while values from 0° to +90° degrees relative to the horizontal with 0° toward the caudal end correspond to a caudal tilt of dorsal cilia. Orientations of cilia were determined after comparing output values using two different MATLAB functions, in addition to a manual calculation, to conclude which value best fit the data.

Quantifications of cilia orientation in respect to the dorsoventral axis, cilia polarity direction and main ciliary beating frequency were then calculated for specific cilia before and after RF photoablation.

The clustering of the fiber and cilia patches (cropped images) was obtained by applying the standard k-means algorithm with $k = 2$ (two clusters). Considering each image $i$ (with $1 \leq i \leq p$) as a vector $x_i \in \mathbb{R}^n$ ($n$ being the number of pixels of each image, and $p$ the number of images), we initialized the cluster centers $c_j$ ($1 \leq j \leq k$) with $c_j = x_j$ and then iterated the classical two-steps k-means procedure until convergence: (1) for each $i$, associate image $i$ to the cluster $j$ such that $\|x_i - c_j\|$ is minimal; (2) for each $j$, recompute the center $c_j$ as the average of all $x_i$ that are associated to cluster $j$.

## Immunohistochemistry

Experiments were done on 3 dpf *Tg(sspo:sspo-GFP;β-actin:Arl13b-GFP)* double transgenic larvae. Larvae were first euthanized with 0.2% Tricaine and then fixed in a solution containing 4% paraformaldehyde solution (PFA), 1% DMSO and 0.3% Triton X-100 in PBS (0.3% PBSTx) at 4 °C overnight. The samples were then washed once with 0.3% PBSTx to remove any traces of the fixation solution. For permeabilization, samples were incubated for 10 min at –20 °C with acetone. Subsequently, samples were washed with 0.3% PBSTx (3x10 min) and blocked in a solution containing 0.1% BSA and 0.3% PBSTx for 2 hr at room temperature. Samples were incubated overnight at 4 °C with glutamylated tubulin (GT335, 1:400, Adipogen) for staining motile cilia, and GFP antibody (AB13970, 1:400, Abcam) to amplify the signals of the Reissner fiber and motile cilia in the primary antibody solution containing 0.1% BSA and 0.3% PBSTx. The next day, samples were washed with 0.3% PBSTx (3x1 hr) and subsequently incubated overnight at 4 °C with the secondary antibodies, Goat anti-Chicken Alexa Fluor Plus 488 (1:500) and Goat anti-Mouse Alexa Fluor 568 (1:500) (Thermofisher Scientific), with DAPI (1:1000) to stain for neuronal markers. After incubation with the secondary antibody, the larvae were washed (0.3% PBSTx, 3x1 hr) and mounted on glass slides in Vectashield (Vector Laboratories, Burlingame, California, USA) to then be imaged using an inverted confocal SP8X White Light Laser Leica microscope with a 20X objective (N.A.=1).

Immunohistochemistry experiments were also done on 3 dpf *Tg(sspo:sspo-GFP)* transgenic larvae to amplify the signal of the Reissner fiber. Larvae were first euthanized with 0.2% Tricaine and then fixed in a solution containing 4% paraformaldehyde solution (PFA) in PBS at 4 °C overnight. The samples were then washed once with PBS to remove any traces of the fixation solution. Samples were blocked in a PBS solution containing 0.5% TritonX, 1% DMSO and 10% NGS at 4 °C overnight. The next day, samples were incubated overnight at 4 °C with GFP antibody (AB13970, 1:400, Abcam) in the primary antibody solution containing 0.5% TritonX, 1% DMSO and 1% NGS. On the third day, samples were washed with in a PBS solution containing 0.5% TritonX and 1% DMSO (3x15 min), and subsequently incubated overnight at 4 °C with the secondary antibody, Goat anti-Chicken Alexa Fluor Plus 488 (1:500) (Thermo Fisher Scientific), with DAPI (1:1000) to stain for neuronal markers. After incubation with the secondary antibody, the larvae were washed in a PBS solution containing 0.5% TritonX and 1% DMSO (3x15 min) and mounted on glass slides in Vectashield (Vector Laboratories, Burlingame, California, USA) to then be imaged using an inverted confocal SP8X White Light Laser Leica microscope with a 20X objective (N.A.=1).

## Acknowledgements

We thank Sophie Nunes-Figueiredo, Monica Dicu and Antoine Arneau for fish care, the ICM Quant imaging facility and Xavier Baudin in the imaging facility of Institut Jacques-Monod for instrument use, scientific and technical assistance. We gratefully thank Prof. Brian Ciruna for the Tg(β\beta-actin:Arl13b-GFP) transgenic line. This work benefited from equipment and services from the core facilities at the Paris Brain Institute (Institut du Cerveau, ICM, Hôpital Pitié-Salpêtrière, Paris, France). We thank all members of the Wyart lab for critical feedback (https://www.wyartlab.org). This work was supported by the HFSP Program Grants RG0063 coordinated by Claire Wyart, in collaboration with Maria Lehtinen, Harvard University and François Gallaire, EPFL.

## Additional information

### Competing interests

Claire Wyart: Reviewing editor, eLife. The other authors declare that no competing interests exist.

### Funding

| Funder | Grant reference number | Author |
| --- | --- | --- |
| Human Frontier Science Program | 2017/RG0063 | Claire Wyart |

| Funder | Grant reference number | Author |
|---|---|---|

The funders had no role in study design, data collection and interpretation, or the decision to submit the work for publication.

## Author contributions

Celine Bellegarda, Conceptualization, Resources, Data curation, Formal analysis, Validation, Investigation, Visualization, Methodology, Writing – original draft; Guillaume Zavard, Data curation; Lionel Moisan, Software, Formal analysis, Supervision, Validation, Visualization; Françoise Brochard-Wyart, Resources, Formal analysis, Supervision, Validation, Investigation; Jean-François Joanny, Supervision, Validation, Investigation; Ryan S Gray, Resources, Writing – review and editing; Yasmine Cantaut-Belarif, Conceptualization, Supervision, Methodology; Claire Wyart, Conceptualization, Resources, Data curation, Software, Formal analysis, Supervision, Funding acquisition, Validation, Investigation, Visualization, Methodology, Writing – original draft, Project administration, Writing – review and editing

## Author ORCIDs

Celine Bellegarda  https://orcid.org/0000-0002-4394-295X
Guillaume Zavard  http://orcid.org/0000-0001-8948-4387
Françoise Brochard-Wyart  http://orcid.org/0000-0001-7455-5875
Jean-François Joanny  http://orcid.org/0000-0001-6966-3222
Ryan S Gray  http://orcid.org/0000-0001-9668-6497
Claire Wyart  https://orcid.org/0000-0002-1668-4975

## Ethics

Animal handling and procedures were validated by the Paris Brain Institute (ICM) and the French National Ethics Committee (Comité National de Reflexion Éthique sur l'Expérimentation Animale; APAFIS # 2018071217081175) in agreement with EU legislation. All experimentswere performed on Danio rerio 3 days old larvae of AB Larvae raised in the same conditions. Experiments were performed at RT on 3 days post fertilization (dpf) larvae based on the protocol of each experiment.

## Decision letter and Author response

Decision letter https://doi.org/10.7554/eLife.86175.sa1
Author response https://doi.org/10.7554/eLife.86175.sa2

# Additional files

## Supplementary files

• MDAR checklist

## Data availability

All code are accessible on GitHub and processed data from imaging and ablation experiments are available here: https://doi.org/10.5061/dryad.573n5tbc2.

The following dataset was generated:

| Author(s) | Year | Dataset title | Dataset URL | Database and Identifier |
|---|---|---|---|---|
| Bellegarda C | 2023 | Data from: The Reissner fiber under tension in vivo shows dynamic interaction with ciliated cells contacting the cerebrospinal fluid | https://doi.org/10.5061/dryad.573n5tbc2 | Dryad Digital Repository, 10.5061/dryad.573n5tbc2 |

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
