## [Editor Report]

This exceptional work substantially advances our understanding of the mechanics of the Reissner's fibre (RF) by performing in-vivo experiments that track and analyze the behavior of the RF when it is cut and the behavior of ciliated cells touching the RF when contact is interrupted. The data is valuable and the conclusions are compelling. The work will be of broad interest to many research communities including developmental neuroscience and cilia biology.

---

## [Decision Letter]

**Decision letter after peer review:**

Thank you for submitting your article "The Reissner fiber under tension in vivo shows dynamic interaction with ciliated cells contacting the cerebrospinal fluid" for consideration by *eLife*. Your article has been reviewed by 3 peer reviewers, including Eva Kanso as Reviewing Editor Reviewer #1, and the evaluation has been overseen by Didier Stainier as the Senior Editor. The following individual involved in the review of your submission has agreed to reveal their identity: Catherina Becker (Reviewer #2).

Essential revisions:

The revised version should fix the typos pointed out by Reviewer #1 and address the suggestions by Reviewer #2.

*Reviewer #1 (Recommendations for the authors):*

I have a couple of small comments on the mechanical model in section 3. The equations will be easier to read if the equation number is separated from the equation. Equation 2 has a typo: in the first fraction, the denominator should be ln(L/r_f), that is the natural logarithm, which appears in the slender-body approximation of the drag coefficient. Please mention explicitly the slender-body approximation employed here.

*Reviewer #2 (Recommendations for the authors):*

I have only two small suggestions:

3A is shown at a different magnification from A1 and A2. This should be adjusted, to match B (B1 and 2) and C (C1 and 2). Also, there is no reference to the scale bar in C in the figure legend.

"Hundreds of thousands" (of beating mono cilia) is a vague number, can this be specified better (both how the estimate was done and what the number was that the estimate yielded)?

---

## [Author Response]

Essential revisions:Reviewer #1 (Recommendations for the authors):I have a couple of small comments on the mechanical model in section 3. The equations will be easier to read if the equation number is separated from the equation.

Agreed and fixed in the revised version.

Equation 2 has a typo: in the first fraction, the denominator should be ln(L/r_f), that is the natural logarithm, which appears in the slender-body approximation of the drag coefficient. Please mention explicitly the slender-body approximation employed here.

Agreed and fixed in the revised version.

Reviewer #2 (Recommendations for the authors):I have only two small suggestions:3A is shown at a different magnification from A1 and A2. This should be adjusted, to match B (B1 and 2) and C (C1 and 2). Also, there is no reference to the scale bar in C in the figure legend.

Agreed and fixed.

"Hundreds of thousands" (of beating mono cilia) is a vague number, can this be specified better (both how the estimate was done and what the number was that the estimate yielded)?

We thank the reviewer for asking for precision. We think our assumption was indeed an overestimation. The central canal is 3.5 mm long and based on EM (Orts-Del-Immagine et al., Current Biology 2020) and our transgenic lines (Sternberg et al., Nature Communications 2019), we estimated the number of cilia to be about 60 per 10 μm along the rostrocaudal axis, which would lead to 60 x 350 ~ 21,000 cilia. This estimation has been corrected in the main text of the revised manuscript.